# UNIVERSAL CLUSTERING BOUNDS

## ABSTRACT

This paper seamlessly integrates several fundamental learning tasks under the umbrella of subspace clustering, namely orthogonal nonnegative matrix factorization, and K-means clustering. Within this framework, we unveil a unified, closed-form solution that elegantly addresses these tasks. Our main theoretical contribution establishes that our deterministic solution achieves perfect accuracy when the data exhibits sufficiently well-defined clusters. Furthermore, the immediate relaxation of our solution yields practical algorithms that not only rival but also surpass the current state-of-the-art in these complex problem domains. This achievement is corroborated by a comprehensive array of experiments conducted on synthetic datasets, as well as on a diverse set of five real-world datasets.

## 1 INTRODUCTION

A series of groundbreaking papers have shown the profound interplay between principal component analysis (PCA), orthogonal nonnegative matrix factorization (ONMF), and the classical K-means clustering problem (Ding & He, 2004; Ding et al., 2005; Li & Ding, 2006). In this paper we demonstrate that these challenges are, in fact, special cases of the more general task known as subspace clustering (SC) (Elhamifar & Vidal, 2013). Through this perspective, we present an overarching description of the closed-form solution to all problems. The main insight behind our solution is that in all these models, the features of the data lie near a subspace whose projection operator encodes the clustering. Our main theoretical result is an exact, universal, and deterministic clustering guarantee that is applicable to all these problems, and does not depend on the data or noise distribution. Intuitively, our findings show that the closed-form solution will be correct whenever the clusters formed by the data are sufficiently separated from one another, and data in each cluster not too scattered, or equivalently, whenever the clusters are well-defined. Stemmed from the connection between these learning tasks, we are able to engineer state-of-the-art algorithms for ONMF and K-means that are based on powerful SC clustering methods. These algorithms may improve on the closed-form solution in some settings (even though our theoretical guarantees do not directly apply to them). Our analytical findings are complemented with an exhaustive series of empirical experiments, meticulously conducted on both synthetic and authentic datasets.

## 2 EVERY MODEL ALL AT ONCE

In this section we will show that PCA, K-means, and ONMF are all special cases of SC. To this end, we first describe the SC model. We will then express it in an intuitive way, and describe how it simplifies to each of the other models.

### SUBSPACE CLUSTERING

Let $\{\mathbf{x}_i\}$ be a collection of $n$ samples lying near the union of $K$ $r$-dimensional subspaces of $\mathbb{R}^m$. Specifically, let $\{\Omega_k\}$ be a partition of $\{1, \ldots, n\}$ indicating the *true* clustering of the samples among the $K < \min(m, n)/r$ subspaces. Let $\mathbf{U}_k \in \mathbb{R}^{m \times r}$ be a basis of the $k^{\text{th}}$ *true* subspace $\mathcal{U}_k$. Suppose

$$\mathbf{x}_i = \sum_{i=1}^{n} \mathbf{U}_k \, \mathbf{v}_i \mathbb{1}_{\{i \in \Omega_k\}} + \mathbf{z}_i, \tag{1}$$

where $\mathbf{v}_i \in \mathbb{R}^r$ is the vector of coefficients of $\mathbf{x}_i$ with respect to the basis $\mathbf{U}_k$, $\mathbb{1}$ denotes the indicator function, and $\mathbf{z}_i \in \mathbb{R}^m$ determines the separation of $\mathbf{x}_i$ from its corresponding subspace, which can

be interpreted as noise. Given $\{\mathbf{x}_i\}$, the goal of SC is to estimate the true clusters $\{\Omega_k\}$ and the true subspaces $\{\mathcal{U}_k\}$.

To express SC in a canonical form where the connection to all other models is evident, first, let $\mathbf{U} := [\mathbf{U}_1, \ldots, \mathbf{U}_K]$ be the concatenation of the bases $\{\mathbf{U}_k\}$. Next, define $\mathbf{V}_k \in \mathbb{R}^{n \times r}$ as the matrix whose $i^{\text{th}}$ row is equal to $\mathbf{v}_i^\top$ if $i \in \Omega_k$, and zero otherwise. Similar to $\mathbf{U}$, let $\mathbf{V} := [\mathbf{V}_1, \ldots, \mathbf{V}_K] \in \mathbb{R}^{n \times Kr}$ be the concatenation of $\{\mathbf{V}_k\}$. Notice that $\mathbf{V}$ encodes the information of the clustering, because the $i^{\text{th}}$ row of $\mathbf{V}$ can only be nonzero in the $k^{\text{th}}$ block of width $r$ if and only if $i \in \Omega_k$. This way, (1) can be written compactly for every $i$ as

$$\mathbf{X} = \mathbf{U}\mathbf{V}^\top + \mathbf{Z}, \tag{2}$$

where $\mathbf{Z} \in \mathbb{R}^{m \times n}$ is the matrix formed with the columns in $\{\mathbf{z}_i\}$. For example, if the $k^{\text{th}}$ block of columns in $\mathbf{X}$ lie in the $k^{\text{th}}$ subspace, then $\mathbf{X}$ would look something like this:

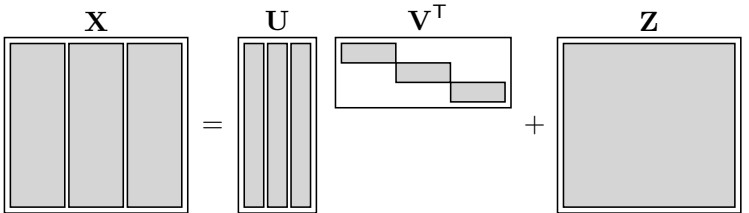

Here, the white and shaded areas represent zero and non-zero entries, respectively.

PCA, ONMF, AND K-MEANS ARE SPECIAL CASES OF SC

From (2), it is evident that:

- **PCA** is the special case of SC where $K = 1$, so that $\mathbf{U} = \mathbf{U}_1 \in \mathbb{R}^{m \times r}$, and $\mathbf{V} = \mathbf{V}_1 \in \mathbb{R}^{n \times r}$. Given $\mathbf{X}$, the goal of PCA is to estimate $\mathbf{U}$, and $\mathbf{V}$ (up to direction and permutation).

- **ONMF** is the special case of SC where $r = 1$, so that each $\mathbf{U}_k$ is a single column, same as $\mathbf{V}_k$, with the additional assumptions that $\mathbf{X}, \mathbf{U} \in \mathbb{R}^{m \times K}$, and $\mathbf{V} \in \mathbb{R}^{n \times K}$ have no negative entries, and that $\mathbf{V}$ is orthogonal. The disjoint support of $\mathbf{V}$ is induced precisely by these two assumptions. Given $\mathbf{X}$, the goal of ONMF is to estimate $\mathbf{U}$ and $\mathbf{V}$ (up to scaling and permutation).

- **K-means** is the special case of SC where $r = 1$ and $\mathbf{v}_i = 1 \; \forall \; i$, so each $\mathbf{U}_k$ is a single column (the cluster center, typically denoted as $\boldsymbol{\mu}_k$), same as $\mathbf{V}_k$. Hence, the $(i, k)^{\text{th}}$ entry of $\mathbf{V} \in \mathbb{R}^{n \times K}$ is either 1 (if the $i^{\text{th}}$ sample corresponds to the $k^{\text{th}}$ cluster), or 0 (otherwise). Given $\mathbf{X}$, the goal of K-means is to estimate the cluster centers $\mathbf{U} = [\mathbf{U}_1, \ldots, \mathbf{U}_K] = [\boldsymbol{\mu}_1, \ldots, \boldsymbol{\mu}_K] \in \mathbb{R}^{m \times K}$ and the clusters $\{\Omega_k\}$ (encoded in $\mathbf{V}$).

## 3 ONE CLOSED-FORM SOLUTION FOR EVERYTHING

Here we describe a unified closed-form solution to these four problems. The key observation is that under each of these models, the features of the data lie near a subspace whose projection operator $\mathbf{P}$ encodes the clustering, which in turn determines $\mathbf{U}$ and $\mathbf{V}$. To see this, observe that the supports of $\{\mathbf{V}_k\}$ are disjoint. So we can obtain orthonormal bases $\{\bar{\mathbf{V}}_k\}$ with the same spans and supports as $\{\mathbf{V}_k\}$, such that $\bar{\mathbf{V}} = [\bar{\mathbf{V}}_1, \ldots, \bar{\mathbf{V}}_k]$ is orthonormal and spans the same subspace as $\mathbf{V}$. Since projection operators are unique, it follows that the projection operator onto $\text{span}(\mathbf{V}) = \text{span}(\bar{\mathbf{V}})$ is

$$\mathbf{P} = \bar{\mathbf{V}}\bar{\mathbf{V}}^\top. \tag{3}$$

From (3) we can see that the columns and rows of $\mathbf{P}$ have the exact same supports as the columns in $\bar{\mathbf{V}}$ (and $\mathbf{V}$), which encode the clustering. For example, the supports of a pair $(\mathbf{V}, \mathbf{P})$, with $\mathbf{V}$ as above, would look like:

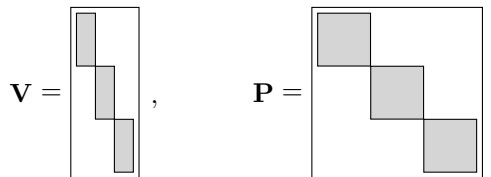

A closed-form solution can be trivially obtained by estimating $\mathbf{P}$ using a singular value decomposition, and a threshold operation to reveal the clustering. More precisely, let $\tilde{\mathbf{V}} \in \mathbb{R}^{n \times Kr}$ be the matrix containing the $Kr$ leading right singular vectors of $\mathbf{X}$. It is clear that even if $\mathbf{Z} = \mathbf{0}$, $\tilde{\mathbf{V}}$ may be an entirely different basis from $\mathbf{V}$, and it may not reveal the desired clustering. However, since projection operators are unique, and $\mathbf{P}$ does encode the clustering, we can estimate $\mathbf{P}$ as $\hat{\mathbf{P}} := \tilde{\mathbf{V}}\tilde{\mathbf{V}}^\mathsf{T} \in \mathbb{R}^{n \times n}$. To estimate the support of $\mathbf{P}$ (and hence the clustering), we can then define $\hat{\mathbf{P}}_\lambda$ as the entry-wise thresholded version of $\hat{\mathbf{P}}$ with entries

$$[\hat{\mathbf{P}}_\lambda]_{ij} := \begin{cases} \hat{\mathbf{P}}_{ij} & \text{if } |\hat{\mathbf{P}}_{ij}| > \lambda \\ 0 & \text{otherwise,} \end{cases}$$

where $\lambda \in [0, 1]$ is a parameter that depends on the positions of the subspaces, and the intra-cluster dispersion level determined by $\{\mathbf{z}_i\}$ (more details below). The cluster estimates, denoted as $\{\hat{\Omega}_k\}$, are the distinct supports (i.e., non-zero patterns) of the columns in $\hat{\mathbf{P}}_\lambda$. The $k^{\text{th}}$ subspace estimate is spanned by the matrix $\hat{\mathbf{U}}_k \in \mathbb{R}^{m \times r}$ formed with the leading left singular vectors of the matrix $\mathbf{X}_k$ formed with the columns assigned to the $k^{\text{th}}$ cluster, i.e., $\{\mathbf{x}_i : i \in \Omega_k\}$. Finally, the $k^{\text{th}}$ coefficient matrix is given by $\hat{\mathbf{V}}_k := \hat{\mathbf{U}}_k^\mathsf{T}\mathbf{X}_k$ (the coefficients of the orthogonal projection of $\mathbf{X}_k$ onto $\text{span}(\hat{\mathbf{U}}_k)$).

Notice that this solution gives appropriate outputs in each case. For PCA we only need to keep the factors $\hat{\mathbf{U}} = \hat{\mathbf{U}}_1 \in \mathbb{R}^{m \times r}$ and $\hat{\mathbf{V}} = \hat{\mathbf{V}}_1 \in \mathbb{R}^{n \times r}$ (recall that PCA is the special case with $K = 1$). In K-means we only need to keep $\hat{\mathbf{U}} = [\hat{\mathbf{U}}_1, \dots, \hat{\mathbf{U}}_K] \in \mathbb{R}^{m \times K}$ and $\{\Omega_k\}$ (recall that K-means is the special case with $r = 1$). In ONMF, due to the nonnegativity constraint (more on this below), we need to keep $\hat{\mathbf{U}} = [|\hat{\mathbf{U}}_1|, \dots, |\hat{\mathbf{U}}_K|] \in \mathbb{R}^{m \times K}$ and $\hat{\mathbf{V}} \in \mathbb{R}^{n \times K}$, whose $k^{\text{th}}$ column is equal to $|\hat{\mathbf{V}}_k|$ in the locations of $\hat{\Omega}_k$, and zero elsewhere.

## 4 Exact, Deterministic, Global Guarantees Everywhere

Our main theoretical result shows that the closed-form clusterings described above are deterministically correct as long as the samples in each cluster are not too scattered relative to the location of the subspaces. We summarize this in the following theorem.

**Theorem 1.** *Let $\delta > 0$ be the gap between the $(Kr)^{\text{th}}$ singular value of $\mathbf{UV}^\mathsf{T}$ and the $(Kr+1)^{\text{th}}$ singular value of $\mathbf{X}$. Suppose*

$$\delta > \sqrt{2^3 Kr}\|\mathbf{Z}\|/\epsilon, \tag{4}$$

*where $\epsilon > 0$ denotes the entry in the support of $\mathbf{P}$ with the smallest absolute value. Then the estimates $\{\hat{\Omega}_k\}$ obtained with $\lambda = \epsilon/2$ are the* true *clusters $\{\Omega_k\}$.*

Intuitively, the $(Kr)^{\text{th}}$ and $(Kr+1)^{\text{th}}$ singular values of $\mathbf{UV}^\mathsf{T}$ and $\mathbf{X}$ can be respectively interpreted as the smallest separation between subspaces, and the largest separation of a sample from its subspace. The condition on $\delta$ essentially requires that the former (distance between subspaces) is large enough relative to the later (noise variance), so that the clusters are discernible from our estimator $\hat{\mathbf{P}}$, and no sample is missclustered. The proof is in Section 6, and it essentially uses the Davis-Kahan $\sin(\boldsymbol{\Theta})$ Theorem (Davis & Kahan, 1970; Stewart & Sun, 1990) to show that the support of $\hat{\mathbf{P}}_\lambda$ will coincide with that of the *true* $\mathbf{P}$ (projection operator onto the span of $\boldsymbol{\Omega}$). For incoherent matrices

$\hat{\mathbf{P}}$, the condition on $\delta$ can be relaxed by a factor of $\mathcal{O}((\mathrm{Kr})^4/\sqrt{n})$ using the $\ell_\infty$ perturbation bound in (Fan et al., 2018) instead.

From our discussion in Section 2, is easy to see that Theorem 1 applies to all SC, PCA, ONMF, and K-means. In the later, $\epsilon$ simplifies to $1/\max_k |\Omega_k|$. To see this, recall that the $(i, k)^{\text{th}}$ entry of $\mathbf{V}$ is equal to 1 if $i \in \Omega_k$, and 0 otherwise. Then the $k^{\text{th}}$ column of its normalized version $\bar{\mathbf{V}}$ takes the value $1/\sqrt{|\Omega_k|}$ if $i \in \Omega_k$, and 0 otherwise. Then the smallest entry in $\mathbf{P} = \bar{\mathbf{V}}\bar{\mathbf{V}}^{\mathsf{T}}$ is equal to $1/\max_k |\Omega_k|$. In general, $\epsilon$ is always larger than zero for subspaces in general position.

As direct consequence of a correct clustering, we obtain the following corollary:

**Corollary 1** (Optimal estimators). *Under the assumptions of Theorem 1, the closed-form estimates $\hat{\mathbf{U}}$ and $\hat{\mathbf{V}}$ are the optimal mean-squared estimators of $\mathbf{U}$ and $\mathbf{V}$.*

Again, notice that Corollary 1 applies to all, SC, PCA, ONMF, and K-means. In some cases it might be desired to obtain estimators that minimize some other loss. Corollary 1 can be trivially generalized to those settings. In the particular case of ONMF, Corollary 1 does not explicitly guarantee the nonnegativity of $\hat{\mathbf{U}}$ nor $\hat{\mathbf{V}}$. However, since $\mathbf{X}_k$ is nonnegative, its leading singular vectors $\hat{\mathbf{U}}_k$ and $\hat{\mathbf{V}}_k$ (or $-\hat{\mathbf{U}}_k$ and $-\hat{\mathbf{V}}_k$) point in the nonnegative orthant (recall that in the ONMF case, $r = 1$, so $\hat{\mathbf{U}}_k$ and $\hat{\mathbf{V}}_k$ only have one column). We thus obtain the following corollary:

**Corollary 2** (ONMF). *In the special case of ONMF, and under the assumptions of Theorem 1, either $(\hat{\mathbf{U}}_k, \hat{\mathbf{V}}_k)$ or $(-\hat{\mathbf{U}}_k, -\hat{\mathbf{V}}_k)$ are nonnegative. In other words, the closed-form estimates $\hat{\mathbf{U}}$ and $\hat{\mathbf{V}}$ satisfy the nonnegative constraint.*

PRACTICAL CONSIDERATIONS

Notice that in general, $\epsilon$ is unknown, and therefore, so is $\lambda$. However, since Theorem 1 guarantees the existence of a parameter $\lambda$ that results in a perfect recovery of (the support of) $\mathbf{P}$, $\lambda$ can be trivially found by searching for the threshold that produces a matrix $\mathbf{P}_\lambda$ with exactly K distinct disjoint supports that cover all the columns and all the rows. One may even re-scale $\hat{\mathbf{P}}$ as in (Ding & He, 2004), so that its entries can be interpreted as a connectivity probability, and use a direct threshold of $0.5$.

However, perfect recovery of $\mathbf{P}$ is a sufficient condition for clustering, convenient for analysis, but by no means necessary in practice. In practice, even if the support of $\hat{\mathbf{P}}_\lambda$ is different than that of $\mathbf{P}$ for every $\lambda \in [0, 1]$, $\hat{\mathbf{P}}$ may still contain enough information to reveal the clustering through a relaxed method. Examples of such methods could be a form of hierarchical clustering that agglomerates samples $(i, j)$ if $[\hat{\mathbf{P}}_\lambda]_{ij} > 0$ as we decrease $\lambda$ from 1 to 0, a direct application of Lloyd's algorithm on $\hat{\mathbf{P}}$, spectral clustering of $\hat{\mathbf{P}}$, or SC variants of the closed-form solution (see Section 5). Our experiments show that this type of approaches lead to perfect clusterings even if the exact support of $\mathbf{P}$ is not perfectly recovered. Our future work will investigate theoretical properties of such strategies, in hopes to find tighter guarantees.

## 5 RELATED WORK

The connection between some of the models under study has been previously pointed out. In particular, (Ding & He, 2004) showed that K-means is equivalent to PCA, and shows that the PCA closed-form solution gives an answer to K-means. Similarly, (Li & Ding, 2006) showed the equivalence between several clustering methods and NMF, in particular the equivalence between K-means and ONMF (Ding et al., 2005). On the other hand, the closed-form solution has been long-used for SC, for example for object tracking (Costeira & Kanade, 1998). Several seminal works have focused on interesting variants of such solution. Examples include (i) the renowned low-rank representation (LRR) algorithm (Liu et al., 2010), which relaxes the rank constraint for the nuclear norm of $\mathbf{P}$, and adds a term to account for outliers and optimizes using an iterative Augmented Lagrange Multiplier method, (ii) the least-squares representation (LSR) (Lu et al., 2012), which adds a Frobenius regularization to overcome higher levels of noise and results in an alternative closed-form solution, or (iii) the block-diagonal representation (BDR) (Feng et al., 2014; Lu et al., 2018), which includes a block-diagonal prior to favor clustering.

To the best of our knowledge, none of these SC techniques have been adapted for K-means and ONMF, where the prevalent methodologies remain variants of Lloyd's algorithm (Lloyd, 1982; Arthur & Vassilvitskii, 2007; Ahmed et al., 2020), or variants of NMF that integrate the orthogonality constraint, and can only guarantee local convergence (Asteris et al., 2015; Fathi Hafshejani & Moaberfard, 2022; Choi, 2008; Yang & Oja, 2010; Li et al., 2007a; Cao et al., 2007; Li et al., 2007b; Chen et al., 2009; Pompili et al., 2014; Del Buono & Pio, 2015; vCopar et al., 2019; He et al., 2020; Li & ZHANG, 2008; Wang & Zhang, 2012; Huang et al., 2012; Li & Ding, 2018; Gan et al., 2021; Chen et al., 2022). In contrast, we give a perfect clustering deterministic guarantee, applicable to all these problems. Moreover, in our experiments we adapt these SC methods to perform K-means and ONMF to obtain state-of-the-art performance in these problems.

## 6 PROOF

To prove Theorem 1 we use the Davis-Kahan $\sin(\mathbf{\Theta})$ Theorem (Davis & Kahan, 1970; Stewart & Sun, 1990) to show that the entries of our estimator $\hat{\mathbf{P}}$ cannot be too different from the entries in $\mathbf{P}$. Specifically, we will show that corresponding entries in these matrices cannot differ by more than $\epsilon/2$. To see this, write

$$\|\mathbf{P} - \hat{\mathbf{P}}\|_F^2 = \|\mathbf{P}\|_F^2 - 2\mathrm{tr}(\mathbf{P}^\mathsf{T}\hat{\mathbf{P}}) + \|\hat{\mathbf{P}}\|_F^2 = 2\mathrm{Kr} - 2\mathrm{tr}(\mathbf{P}^\mathsf{T}\hat{\mathbf{P}})$$
$$= 2\mathrm{Kr} - 2\mathrm{tr}(\bar{\mathbf{V}}\bar{\mathbf{V}}^\mathsf{T}\hat{\mathbf{V}}\hat{\mathbf{V}}^\mathsf{T}) = 2(\mathrm{Kr} - \|\bar{\mathbf{V}}^\mathsf{T}\hat{\mathbf{V}}\|_F^2)$$
$$=: 2(\mathrm{Kr} - \cos^2(\mathbf{\Theta})) = 2\sin^2(\mathbf{\Theta}) \le 2\|\mathbf{Z}\|_F^2/\delta^2,$$

where the last inequality follows directly by the Davis-Kahan $\sin(\mathbf{\Theta})$ Theorem (Davis & Kahan, 1970; Stewart & Sun, 1990). Then

$$\|\mathbf{P} - \hat{\mathbf{P}}\|_\infty \le \|\mathbf{P} - \hat{\mathbf{P}}\|_F \le \frac{\sqrt{2}\|\mathbf{Z}\|_F}{\delta} \le \frac{\sqrt{2\mathrm{Kr}}\|\mathbf{Z}\|}{\delta}.$$

Substituting $\delta$ from (4), we see that

$$\|\mathbf{P} - \hat{\mathbf{P}}\|_\infty \le \frac{\sqrt{2\mathrm{Kr}}\|\mathbf{Z}\|}{\delta} < \frac{\sqrt{2\mathrm{Kr}}\|\mathbf{Z}\|\epsilon}{\sqrt{2^3\mathrm{Kr}}\|\mathbf{Z}\|} = \frac{\epsilon}{2}.$$

This implies that the difference of any two entries in $\mathbf{P}$ and $\hat{\mathbf{P}}$ is bounded as

$$-\frac{\epsilon}{2} < \mathbf{P}_{ij} - \hat{\mathbf{P}}_{ij} < \frac{\epsilon}{2} \qquad \forall i,j. \tag{5}$$

On the other hand, from (3) and the definitions of $\mathbf{V}$ and $\bar{\mathbf{V}}$, we can see that the entries of $\mathbf{P}$ are

$$\mathbf{P}_{ij} := \begin{cases} \mathbf{P}_{ij} & \text{if } i,j \in \Omega_k \\ 0 & \text{otherwise.} \end{cases} \tag{6}$$

Plugging (6) in the second inequality of (5), we see that if $i,j \in \Omega_k$,

$$\hat{\mathbf{P}}_{ij} > \mathbf{P}_{ij} - \frac{\epsilon}{2} \ge \epsilon - \frac{\epsilon}{2} = \frac{\epsilon}{2}.$$

Similarly, plugging (6) in the first inequality of (5), we see that if $i$ and $j$ are not in the same $\Omega_k$,

$$\hat{\mathbf{P}}_{ij} < 0 + \frac{\epsilon}{2} = \frac{\epsilon}{2}.$$

Taking $\lambda = \frac{\epsilon}{2}$, we see that after thresholding,

$$[\hat{\mathbf{P}}_\lambda]_{ij} = \begin{cases} \hat{\mathbf{P}}_{ij} > \frac{\epsilon}{2} & \text{if } i,j \in \Omega_k \\ 0 & \text{otherwise.} \end{cases}$$

We thus see that the supports of $\mathbf{P}$ and $\hat{\mathbf{P}}_\lambda$ are identical, which concludes the proof.

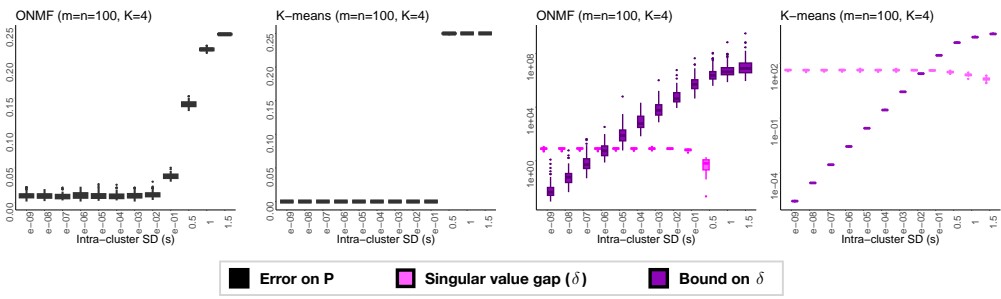

**Figure 1: Left:** Error on the estimated support of $\mathbf{P}$. **Right:** Singular value gap $\delta$, and its bound (4). These results verify Theorem 1, showing that our clustering solution is exact if $\delta$ satisfies (4).

## 7 EXPERIMENTS

This section presents a thorough series of experiments with two main purposes: (i) verify the deterministic exact clustering guarantee in Theorem 1, and (ii) demonstrate the practical performance of state-of-the-art SC algorithms in the K-means and ONMF settings. In these experiments we generated data according to (1), populating the entries of each $\mathbf{U}_k \in \mathbb{R}^{m \times r}$ with i.i.d. standard normal random variables. According to our discussion in Section 2, to emulate the K-means setting, we set $r = 1$ and $\mathbf{v}_i = 1 \, \forall \, i$. To emulate the ONMF setting, we set $r = 1$, and populated $\mathbf{v}_i$ with i.i.d. standard normal random variables. In all cases, we populated the entries of each $\mathbf{z}_i$ with i.i.d normal with zero mean and variance $s^2$. The n generated samples are evenly distributed among the K clusters $\{\Omega_k\}$ (up to rounding error).

### 7.1 VALIDATING THEORY

In our first experiment we verify Theorem 1. To this end, we measured (a) the singular value gap $\delta$, (b) the bound in the right hand side of (4), and (c) the error on the estimated support of $\mathbf{P}$, i.e., the average number of distinct nonzero entries in $\mathbf{P}$ and $\hat{\mathbf{P}}_\lambda$, as a function of $s^2$, which represents the intra-cluster variance, related to $\|\mathbf{Z}\|$ in (4). The results of 100 independent trials for each value of s are in Figure 1, where we used a fixed $\lambda = s$, with $m = n = 100$ and $K = 5$. Theorem 1 shows that $\mathbf{P}$ can be perfectly recovered if $\delta$ satisfies the condition in (4) (whenever the magenta line is above the purple line). Figure 1 verifies this result, and suggests that the bound in (4) can be tightened, at least stochastically, if not deterministically, as the support of $\mathbf{P}$ can still be perfectly recovered from $\hat{\mathbf{P}}_\lambda$ when $\delta$ is outside the bound (see the range $s \in (10^{-6}, 10^{-2})$ in the ONMF experiments, where $\delta$ is outside the Theorem's bound, yet the support of $\mathbf{P}$, and hence the clustering, can be perfectly recovered). We point out that recovering the exact support of $\mathbf{P}$ (as required by Theorem 1) is a sufficient, but by no means necessary condition for clustering. Our next experiments show that using simple relaxations one can still perfectly recover the true clustering from $\hat{\mathbf{P}}$ even if the support of $\hat{\mathbf{P}}_\lambda$ is not identical to that of $\mathbf{P}$.

### 7.2 PRACTICAL RELAXATION

In our second experiments we use a simple relaxation of the closed-form solution that runs spectral clustering (Ng et al., 2001) on $\hat{\mathbf{P}}$ (CF+SpC). This approach can be used in cases where the support of $\mathbf{P}$ cannot be perfectly estimated directly from $\mathbf{P}_\lambda$. These experiments aim to show that even in these cases, $\hat{\mathbf{P}}$ still encodes enough information to reveal the clustering with typical methods that minimize distortion (like spectral clustering).

**Baselines.** For comparison, we use a comprehensive mix of classical and state-of-the-art K-means, ONMF, and SC methods: Lloyd's algorithm (Lloyd, 1982), K-means++ (Arthur & Vassilvitskii, 2007), alternating (AONMF) (Pompili et al., 2014), hierarchical alternating least-squares (HALS) (Shiga et al., 2016), orthogonal nonnegative matrix T-factorizations (ONMTF) (Ding et al., 2006), multiplicative updates (MU) (Lee & Seung, 2000), block-diagonal representation (BDR) (Lu et al., 2018), low-rank representation (LRR) (Liu et al., 2010), least-squares representation (LSR) (Lu et al., 2012), and sparse subspace clustering (SSC) (Elhamifar & Vidal, 2013). We used our own

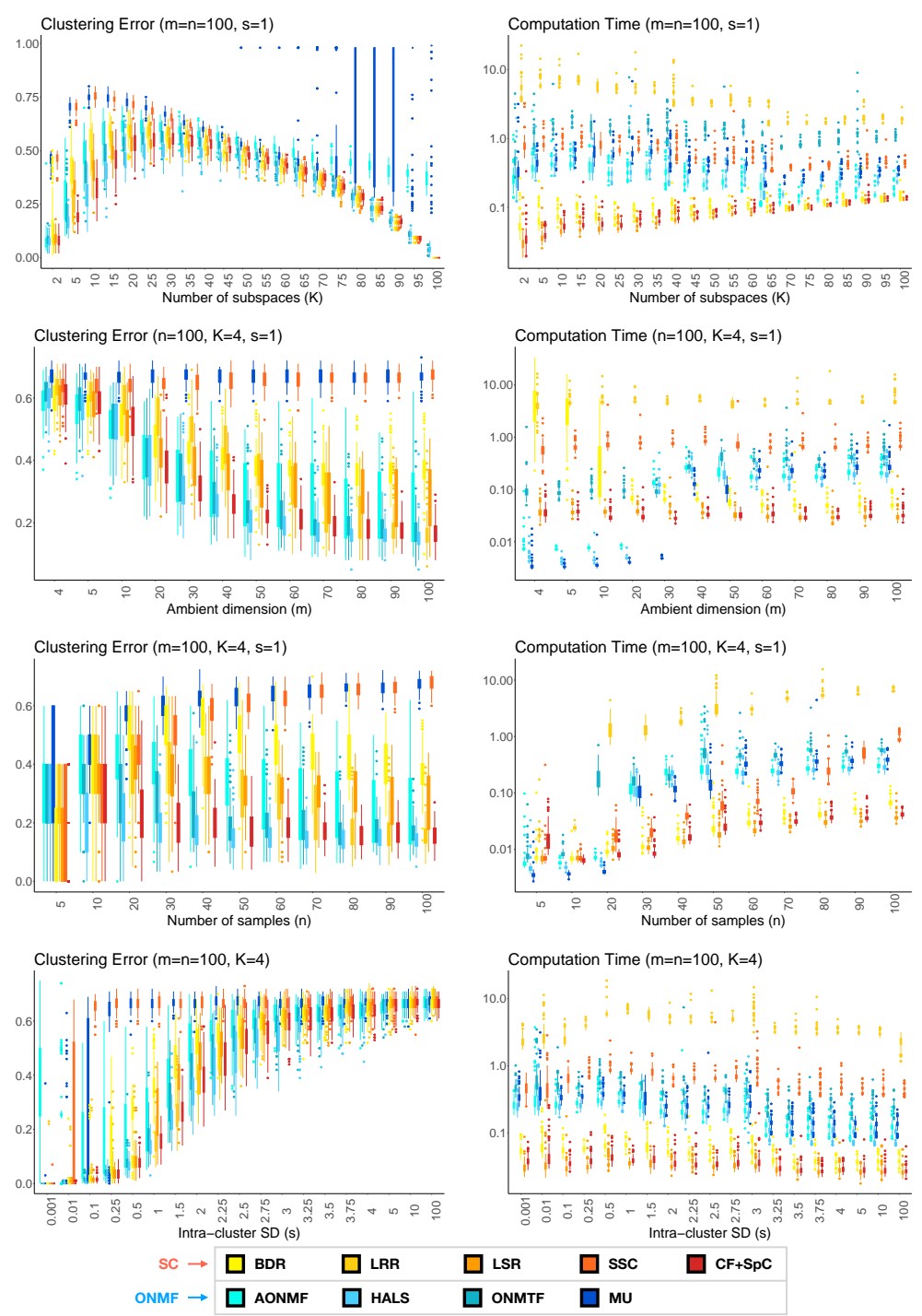

**Figure 2:** Clustering error and computation time as a function of different parameters under the ONMF setting. Our SC closed-form relaxation (CF+SpC) generally matches and often improves the accuracy of the state-of-the-art, but it is orders of magnitude faster.

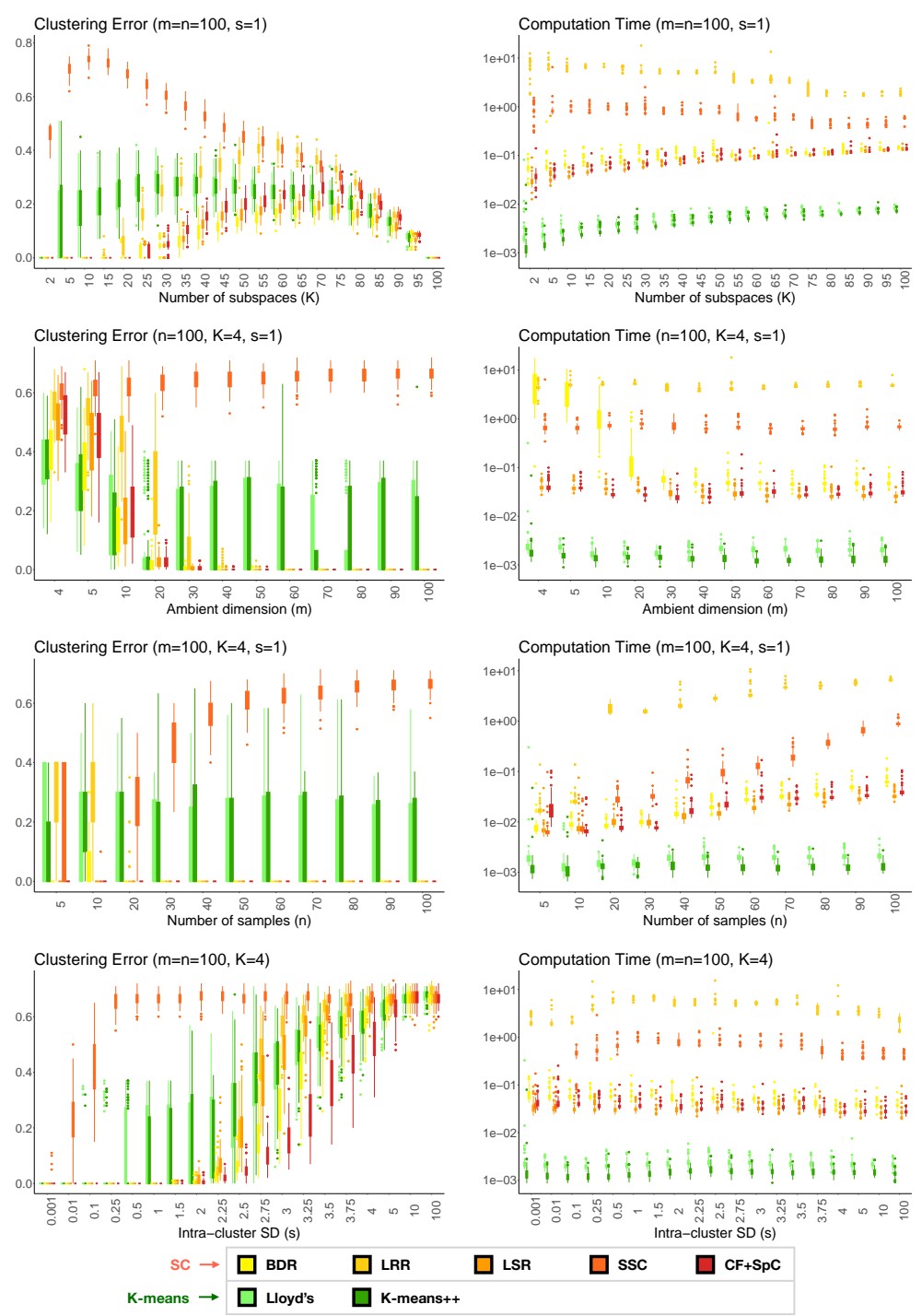

**Figure 3:** Clustering error and computation time as a function of different parameters under the K-means setting. Our SC closed-form relaxation (CF+SpC) generally matches and often improves the accuracy of the state-of-the-art, but it is orders of magnitude faster.

| Algorithm | Buenrostro6 | Buenrostro7 | Larry | Nestorowa | Stassen |
|---|---|---|---|---|---|
| BDR | 0.4435 | 0.4031 | 0.6043 | 0.4042 | 0.696 |
| LSR | 0.3669 | 0.35 | 0.5816 | 0.4542 | **0.273** |
| AONMF | 0.3803 | 0.2271 | 0.5779 | 0.4318 | 0.485 |
| HALS | 0.4353 | 0.44 | **0.5607** | 0.3828 | 0.832 |
| ONMTF | 0.4581 | 0.4707 | 0.5682 | 0.4661 | 0.635 |
| MU | 0.4259 | 0.4433 | 0.5844 | **0.3417** | 0.615 |
| **CF+SpC (this paper)** | **0.2425** | **0.1974** | 0.5751 | 0.4505 | 0.6580 |

**Table 1:** Clustering error in several real datasets related to single-cell sequencing. The best result is highlighted in bold. our approach significantly outperforms the state-of-the-art in several datasets. In other cases, classical ONMF methods are the best, and in other cases, other SC methods are the best. The unified framework presented by this paper enables the use of all these methods in these tasks.

implementation of Lloyd's and K-means++. All ONMF implementations were obtained from the widely used library (Kasai, 2017), last updated in 2022. All SC implementations were obtained from their respective authors.

**Evaluation.** As performance metric we use the number of missclassified samples after using the Hungarian algorithm (Kuhn, 1955) to find the best labels assignment.

We compare these methods as a function of several parameters of the problem, namely, the number of clusters $K$, the ambient dimension $m$, the number of samples $n$, and the intra-cluster variance $s^2$. The results of 100 trials for each parameter value are in Figures 2 and 3, where SC, ONMF, and K-means algorithms are depicted with warm, cold, and green colors, respectively. The main takeaway from these experiments is that our relaxation of the SC closed-form solution (CF+SpC) generally matches and often improves the accuracy of the state-of-the-art in both ONMF and K-means, but it is orders of magnitude faster.

## 7.3 REAL DATA EXPERIMENTS

We complement our experiments using five real datasets on single-cell sequencing, obtained from the Gene Expression Omnibus (Edgar et al., 2002), namely the Buenrostro 2018 data with 6 and 7 classes, the Larry dataset with 5 classes, th Nestorowa dataset with 3 classes, and the Stassen data with 10 classes. As the name suggests, these data contain gene activation levels of a collection of cells of different types. The goal is to cluster the cells. Some of the methods we used for our simulations were infeasible due to the large scale of these datasets. The results are summarized in Table 1, where we can see that our approach significantly outperforms the state-of-the-art in several datasets. In other cases, classical ONMF methods are the best, and in other cases, other SC methods are the best. The unified framework presented by this paper enables the use of all these methods in these tasks.

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
