# OpenReview forum: "Universal Clustering Bounds"
_ICLR.cc/2024/Conference — Submitted to ICLR 2024_

### Official Review · Reviewer_qhZS · 2023-10-28

**Soundness:** 3 good
**Presentation:** 2 fair
**Contribution:** 2 fair
**Rating:** 5
**Confidence:** 4

**Summary:**

This paper presents a unified perspective on three fundamental problems: Principal Component Analysis (PCA), Orthogonal Nonnegative Matrix Factorization (ONMF), and k-means clustering by viewing each problem as a special case of the k subspace clustering problem (KSC).

To address the KSC problem, the authors propose an algorithm that utilizes the Singular Value Decomposition (SVD) of the input matrix and incorporates a threshold filter governed by the hyperparameter $\lambda$. Subsequently, they provide a theoretical proof of the algorithm's effectiveness for point sets located near the union of k subspaces.

In order to validate their algorithm, the authors conduct experiments that compare its performance not only on the general KSC problem but also on the aforementioned specific cases.

**Strengths:**

- The paper unifies some classic problems in a concise way. Although some connections have been founded, the summary of this paper is clear and easy to follow.
- The algorithm it proposed is simple and easy to implement.
- The theoretical result the paper established connects the norm of $ ||Z|| $and the gap between the singular value of $||X||$ and its approximation. This finding is new to me.

**Weaknesses:**

- The assumption that points lie near the subspace can be strong in practical scenarios. A more general modeling approach involves replacing subspaces with flats, commonly known as projective clustering. In projective clustering, the inherent cluster does not necessarily need to contain the origin point. Additionally, It appears to me that small $||Z||$ is impossible when m and n are large.
- SVD is time-consuming. In the experiments, both m and n are very small. However, as m and n increase, the limitation imposed by the time complexity of SVD becomes more apparent. In the context of special cases, numerous efficient algorithms (e.g., fast PCA) have been developed while this paper is short of such content. In my opinion, if this work aims to propose practical algorithms, it should include a detailed or outlined strategy for acceleration.
- The lack of discussions regarding the norm of $Z$ (see questions below)

**Questions:**

- The theoretical result in this work assumes well-defined clusters, quantitatively characterized by having a small $\||Z\||$. However, there is no explicit discussion about this aspect in the paper.
    - $Z$ is thought as the noise, hence it is natural to assume that it follows some certain distribution. What about the implications of this assumption on the value of $\||Z\||$?
    - What is the rate of increase of $\||Z\||$ when n and m increase? As for the assumption in theorem 1, how do values of n and m affect its feasibility, will larger n or m make it harder to hold?
- Does the computation time reported in the experiment include the decision time for determining hyperparameter $\lambda$?
- The main body of the paper lacks an optimization point of view of the problem. That is, the goal is to minimize $\ell := \||X - UV^T\||$ and define $OPT$ ($\||Z\||$) as the minimal value of $\ell$. Since $OPT$  is unknown beforehand, can you explain what the output of the closed-form solution would be if $OPT$  is not small enough to meet the assumption in theorem 1?

---

### Official Review · Reviewer_B34v · 2023-10-31

**Soundness:** 2 fair
**Presentation:** 2 fair
**Contribution:** 2 fair
**Rating:** 3
**Confidence:** 3

**Summary:**

The Clustering problem has received significant attention in the field of unsupervised learning. Based on the observation that  features of an input clustering dataset often exhibit proximity to a subspace, this paper gives a deterministic method for obtaining universal clustering bounds through singular value decomposition. The authors also show that, under certain data distribution assumptions, i.e., the data points within the same optimal cluster are closer to each other than other data points outside, a deterministic optimal error bound can be established. Empirical results show that the proposed method achieves better performances compared with other state-of-the-art clustering methods.

**Strengths:**

The proposed method gives a deterministic error bound if the data points of a given dataset are "well separated"

**Weaknesses:**

The idea behind the data distribution assumption proposed in this paper has also been widely used in Lloyd's type methods and approximation algorithms design for $k$-means clustering. Such as in [1]-[2], an $\alpha$-perturbation resilient assumption were introduced. They assume that for each optimal cluster, the data points within it are closer to each other than to the data points in other optimal clusters, which is very similar to the "not too scattered" distribution proposed in this paper. However, this paper did not give detailed analysis of time complexity and memory usage of the proposed method. It is unclear to me that whether the proposed method can be generalized to handle large-scale datasets since there already exists massively parallel $k$-means clustering method [3] that could be used to handle large-scale datasets with an optimal guarantee on clustering cost.

The experimental parts do not give detailed descriptions about the data sizes of the real-world datasets used in this paper.

[1] Cohen-Addad V, Schwiegelshohn C. On the local structure of stable clustering instances[C]//2017 IEEE 58th Annual Symposium on Foundations of Computer Science (FOCS). IEEE, 2017: 49-60.

[2] Angelidakis H, Makarychev K, Makarychev Y. Algorithms for stable and perturbation-resilient problems[C]//Proceedings of the 49th Annual ACM SIGACT Symposium on Theory of Computing. 2017: 438-451.

[3] Cohen-Addad V, Mirrokni V, Zhong P. Massively Parallel $k$-Means Clustering for Perturbation Resilient Instances[C]//International Conference on Machine Learning. PMLR, 2022: 4180-4201.

**Questions:**

Could the authors provide detailed analysis of the time and space complexities for the proposed method.

Could the authors compare the data distribution assumptions made in this paper for finding the optimal guarantee of the solution, with the perturbation-resilient assumptions used in approximation design for $k$-clustering problems.

**Details Of Ethics Concerns:**

I don't think this paper has any ethics concerns

---

### Official Review · Reviewer_Eiut · 2023-10-31

**Soundness:** 3 good
**Presentation:** 1 poor
**Contribution:** 1 poor
**Rating:** 3
**Confidence:** 4

**Summary:**

The paper suggests that if the dataset contains points near the union of K r-dimensional subspaces, such points can be clustered using SVD, followed by truncation of small values. This is done by noting that such a data matrix X can be written as:
X = UV^T + Z, where U is a column matrix, V is a matrix with block diagonal representation, and Z has small entries (if the points lie perfectly on the subspaces, then Z=0). A quantitative statement in support of the above idea is given, and experimental results for supporting the suggested algorithm are given.

**Strengths:**

Subspace clustering is a relevant problem.

**Weaknesses:**

- Writing X = UV^T + Z and then interpreting this as instances of PCA, k-means, ONMF (setting different parameters) is a trivial observation and cannot be considered the main contribution. SVD, followed by truncation, also cannot be considered a novel contribution. Overall, the paper lacks new ideas.
- The paper is not well written. Some questions remain unanswered while reading the paper, making the paper lack clarity:
   - What are U, V, and Z for a given X. Are these matrices unique? Does Z satisfy some conditions to make them unique?
   - What does "smallest separation between subspace" mean?
   - Can the condition of Theorem-1 be tested? If so, how, and if not, what is the utility of this theorem?
- I do not see the point of validating the theory section (section 7.1). If you are giving a Theorem, why is there a need to check if it holds using simulated data?

Overall, the quality of the write-up needs to be improved before this paper can be considered for publication.

**Questions:**

Some of the questions are mentioned in the weakness section.

---

### Official Review · Reviewer_Wvoi · 2023-11-01

**Soundness:** 2 fair
**Presentation:** 3 good
**Contribution:** 1 poor
**Rating:** 3
**Confidence:** 4

**Summary:**

This paper considers the three fundamental clustering and learning tasks: PCA, ONMF, and K-means as a whole. The authors show that all of these three problems can be written in a unified single expression, and a closed form solution based on singular value decomposition is obtained. Theoretical analysis shows that under a sufficiently high SNR, the closer form solution achieves perfect recovery of the clusters. Experiments confirms their theory, and shows that perfect recovery could be still possible even if the closed-form solution fails

**Strengths:**

Overall, this paper has the following list of strengths

- Well-written and easy to follow
- A single unified expression of the three problems (PCA, ONMF, and K-means) is provided
- A closer-form solution is proposed for perfect (exact) recovery of the clusters
- A theoretical analysis is provided to guarantee the exact recovery when the SNR is sufficiently high, which seems to be technically correct.
- Experiments confirms the existence of a threshold below which exact recovery is achieved by the closed form
- Experiments shows that exact recovery is still possible even when the closed form fails, which is achieved by applying spectral clustering to the new embedding formed by SVD

**Weaknesses:**

**Novelty and contribution**

My first concern is on the lack of novelty and the contribution is unclear, because of the following reasons:
1. The derivation of the unified expression i.e. Eq. (2) in the paper for the three problems (PCA, ONMF, and K-means) seems to be a good idea, but essentially they are different realizations of the low-rank approximation in a form of SVD, so the novelty here is very limited (no criticism on the methodology but just want to say the proposal of this unified expression may not be novel by itself). Also, the equivalence of these three problems has already been observed for a long while, as pointed out by the authors as well in the related work section.
2. The proposed closed-form solution essentially consists of two steps: an SVD step + a refinement step. Again, the novelty here is limited, since using SVD for recovering the low-rank component from its noise-perturbed observation is an extremely common technique, especially it is the standard approach for PCA. Even if the refinement step could be something new (I am not sure but if this is true, would be good to point out in the paper), this part has not been fully evaluated as the algorithm (CF+SpC) tested in the experimental section does not included this step. As a result, it is unclear to me what should be the main novelty of this algorithm.
3. The theoretical analysis is a plus, while its novelty is also limited. Since such an spectral analysis on the low-rank matrix with noise perturbation by using Davis-Kahan theorem is standard, see e.g. Theorem 7 of [1]. So it is good to have a performance guarantee but the analysis itself may not be novel.
4. As a side note, the novelty and contribution on the theoretical analysis could be increased by conducting a more sharp analysis on the condition of $\delta$  (possibly by using $l_\infty$-norm as pointed out by the authors as well), and empirically demonstrated the sharpness (something like Figure 1 in the paper)

Given the above, I hope the author could clarify the main contributions in the revised manuscript.



**Experiments**

My second concern is on the experiments: the proposed method (CF+SpC) tested in experiments basically performs SVD and then apply spectral clustering on the top singular vectors. As a result, I would expect such a method works well since the top singular vectors exhibits a higher SNR than the original data embedding i.e. $x_i$. However, other approaches used for comparison do not apply on the singular vectors as embedding and thus they would definitely performs worse than CF+SpC. Therefore, I would argue that this is an unfair comparison, especially given that the authors claimed CF+SpC achieves SOTA. Here, since the goal is to demonstrate that exact recovery is still possible even if $\hat{P}_\lambda$ fails to directly recover the clusters, it is reasonable to just apply those different clustering approaches on the rows of $\hat{P}$ and see if most of them still achieve exact recovery in this case.

Also, similar to my second argument in the first section, since the refinement step is not included in the experiments when testing on real datasets, I am interested in the performance of the vanilla closed form algorithm with the refinement step but discard the spectral clustering part. Especially, I would like to see when can we find a disjoint support of $\hat{P}_\lambda$, and even if a disjoint support does not exist, would that be possible to find a subset in which a disjoint support can be still achieved, and then partial recovery is still possble? In the current version of this paper, the refinement step is proposed as an important pillar of the algorithm, while it has not been fully tested and empirically evaluated yet, which prevent us from applying it in real scenarios.

**Minor comments**

- It might be worthy to have a summarization on the closed-form algorithm, or an algorithm block. In particular, some important details mentioned in the 'practical considerations' paragraph on page 4 may need to be included, e.g. the tricks for finding a proper choice of $\lambda$
- When introducing the two matrices $V$ and $\bar{V}$, would that be better to just express them as $\bar{V} = V O$ for some unknown $O$? Then the reason for using $P$ for clustering becomes clear.

[1] Von Luxburg, Ulrike. "A tutorial on spectral clustering." Statistics and computing 17 (2007): 395-416.

**Questions:**

As mentioned in the weakness section above. I hope the authors could clarify my major concerns on the novelty of this paper, clearly state the contributions, and address my questions on the experiments. Also, let me know if I missed anything.

---

### Meta-Review · Area_Chair_a8ZL · 2023-12-05

**Metareview:**

This paper makes connections between some common clustering methods. When the SNR is high, the proposed closed-form solution exactly recovers true clusters. The reviewers provided a detailed discussion of the strengths and weaknesses. A strength was the presentation of the connections between the clustering methods. The main weaknesses were around novelty and contributions. There were also substantive concerns about the alignment between the proposed method and experiments used to test the method. The authors did not respond to the reviewer comments in the response period.

**Justification For Why Not Higher Score:**

The reviewers raised concerns about the novelty and experiments that were not responded to by the authors.

**Justification For Why Not Lower Score:**

N/A

---

### Decision · Program_Chairs · 2024-01-16

Reject